# Combination of Ecological Engineering Procedures Applied to Morphological Stabilization of Estuarine Banks after Dredging

**Luís Filipe Sanches Fernandes** [1,*], **António Augusto Sampaio Pinto** [2],
**Daniela Patrícia Salgado Terêncio** [1], **Fernando António Leal Pacheco** [3] and
**Rui Manuel Vitor Cortes** [4]

[1] Centre for the Research and Technology of Agro-Environment and Biological Sciences, Engineering Department University of Trás-os-Montes e Alto Douro, Ap 1013, 5001-801 Vila Real, Portugal; dterencio@utad.pt

[2] Hydraulics, Water Resources and Environment Division, Civil Engineering Department, Faculty of Engineering, University of Porto-Dr. Roberto Frias street, s/n, 4200-465 Porto, Portugal; antonio.pinto@fe.up.pt

[3] Chemistry Research Centre, University of Trás-os-Montes and Alto Douro, Ap. 1013, 5001-801 Vila Real, Portugal; fpacheco@utad.pt

[4] Centre for the Research and Technology of Agro-Environment and Biological Sciences, Forest Sciences and Landscape Architecture Department, University of Trás-os-Montes e Alto Douro, Ap 1013, 5001-801 Vila Real, Portugal; rcortes@utad.pt

[*] Correspondence: lfilipe@utad.pt; Tel.: +351-259-350-397

**Abstract:** Gravel extraction and upstream damming caused profound effects on the estuary of the Lima river (NW Portugal) which was reflected by the collapse of banks, leading further to the destruction of riparian vegetation. This led to consequences such as a progressive negative impact on the preservation of salt marshes over several decades of this protected area, which continued even after the cessation of extraction activities. In this work, we present a restoration project combining civil engineering with soft soil engineering procedures and revegetation, along with two distinct segments, and follow the recovery process. The main intention of the study is to promote hydraulic roughness in order to dissipate energy from peak flows and tides, increasing accretion and indirectly the stimulation of plant succession and salt marsh recovery. We are able to observe that the built structures (an interconnected system of groynes, deflectors and rip-rap/gabion mattress) allowed the erosion process to be detained. However, they did not allow as much sediment as expected to be trapped. The colonization of species (plants) in brackish and tidal water was a difficulty posed by this project. A more extensive restoration of all estuarine areas and river mouths, namely to overcome the sediment deficit, will require proper land-use management at the catchment scale instead of local actions.

**Keywords:** riverbank erosion; restoration; bank stabilisation; vegetation revetment

## 1. Introduction

Coastal salt marshes are ecosystems of great ecological and economic value since they provide habitats and breeding areas for many animal species. They play a crucial role in the food chain, in the quality control of the environment and in the sedimentation dynamics in estuarine systems [1,2]. However, around the world, erosive processes related to dredging activity for navigation or gravel extraction represent essential factors for the loss of salt marshes through erosion under the living root

zones caused by flowing drainage water. This leads to the overhang of marsh vegetation growing over the banks [3,4]. In addition to the positive benefits of accreting sediment, vegetated marshes effectively dissipate wave energy [5], decreasing the impact of turbulence, which can be crucial in entraining sediment, as shown in [6,7]. The eroding process resulting from the degradation of these ecosystems was described by Castillo's group [8] in other marshes of the Iberian Peninsula, leading to the formation of vertical slopes (usually concave in their lower part), the appearance of mass-movement phenomena and the detachment of blocks of the substrate. The horizontal erosion of these slopes typically begins with the undermining of the lower part, just below the zone of live roots. This is followed by the detachment of substrate blocks from the upper part of the slope, detaching the plants from their roots. Of course, navigational conditions in this estuary (mainly for tourism and fisheries) induce another disruptive factor in acceleration erosion because of wave energy [8,9], but also because channel incisions close to the banks are also observed in order to overcome unfavorable depths.

Ecological engineering has been increasingly used in order to stabilize river banks [10–16], but this is still very uncommon in estuarine areas in Portugal. Some other studies supported the use of hydrodynamic models to predict bank stability under flow conditions [17–20]; in addition, these are essential for the identification of adequate vegetation to improve restoration processes [21–24]. In Portugal, there has also been increasing attention paid to soil engineering techniques for the control of fluvial erosion and for the settlement of riparian galleries in physically disturbed streams [25–28].

The Lima Estuary is subject to many detrimental human impacts. It is the recipient of point pollution originating from the town of Viana do Castelo, an important pulp-mill factory and non-point pollution from agriculture. An input of persistent organic pollutants to the estuarine water and nutrients has led to eutrophication processes in the lower part of the estuary [29]. We also call for attention to be paid to the disturbance by boat navigation and the inherent introduction of fuel and paint residuals into the estuarine system. In spite of the multiple human impacts, the main purpose of this study is to stabilize the river banks along the estuary of the Lima river, as a result of decades of unregulated gravel extraction, causing profound effects on the bank morphology and the destruction of riparian vegetation, either due to tidal action or situations of peak flow. Over the past three decades, there has been a progressive bank cutting leading to substantial marsh losses, also affecting recreational activities. Moreover, the present work shows the implementation of a restoration project aimed at the natural reposition of salt marshes. This is an innovative procedure that has not been used before in Portugal and attempts to provide the necessary conditions that may drive the restoration of the lost wetlands (and the protection of the existing ones) by reverting and supporting river banks that surround previous salt marsh areas, which were washed away after the collapse of the protecting banks. The basis of this design was a combination of civil engineering with soil engineering procedures that not only secure river banks against further erosion but may also increase the patterns of sediment deposition. The biophysical recovery and protection processes were carried out along two distinct segments in the right bank (the nearby Cardielos and Portuzelo villages), both in the estuarine zone of the Lima river. We must emphasize that the downstream part of this river, including the estuary, is included in a protected area of Nature 2000, associated with the preservation of wetlands and riparian layers. Therefore, this work is crucial to defining the procedures for more extensive action. We hypothesize that, by increasing the hydraulic roughness along the banks, we may increase the sedimentation rate along the banks and aid the recolonization process.

## 2. Materials and Methods

### 2.1. Study Area

The Lima river catchment is shared between Portugal and Spain, and the run-off average flowing into the estuary is 3298 hm$^3$, whereas 1598 hm$^3$ corresponds to the Portuguese part (which includes a near 35 km length, with an average slope of 0.1%). The downstream part of this river represents a transition between a narrow and steep valley towards a progressive gentle slope (0.024%) along

with a shallow-vee valley form and finally a large floodplain. The average annual precipitation in this hydrographic basin is high (1444 mm) but averages 2745 mm in some sub-basins. It has a very humid climate and is a hydrographic basin with an excess of water availability throughout the year, with water shortages in the summer months [30,31]. These conditions favor the occurrence of frequent floods in the downstream areas of the main catchment. Impacts on water quality are relatively low since we observe a dominant land use of forest stands (eucalyptus and pine trees) and, in the lower parts of the valley, extensive agriculture characterized by small patches of vineyards, orchards and grasslands with cattle breeding. The areas prioritized for the rehabilitation projects of Cardielos and Portuzelo, both on the right bank, were a consequence of the demands of the municipalities due to the loss of recreational grounds and the increasing pressure on multiple infrastructures (marginal roads, sports and leisure equipment, etc.), but also to protect a layer of marshes in the neighborhood.

## 2.2. Disturbance Factors

The intense dredging related to gravel extraction over nearly three decades in the lower segment of the Lima river has led to a complete change in the morphological character of the river mouth, with the main current flow and thalweg being relocated towards the right margin derived from the intense withdrawal of sediments. Moreover, the sedimentary supply dramatically decreased after 1992 due to river regulation (two tail-race dams were built upstream, Alto Lindoso and Touvedo; the former is the second-most important hydropower system in Portugal, with a dam of 110 m in height and a reservoir capacity with 347.8 hm$^3$, with a maximum area of 1072 ha, whereas the Touvedo dam, 7 km below Alto Lindoso, has a height of 43 m, and the reservoir covers 172 ha. Bank instability is the direct consequence of the deepening of the river channel and the exceedance of the critical height of the river banks, which led to its subsequent collapse. We can see in Figure 1 that the comparison of the studied area (upper part of the estuary) between 1965 and 2010 shows the intense sediment loss in this period and the transformation of a braided channel into a progressive linearization of the river banks, resulting in a river with a significantly higher stream power. Damming is known to affect the entire downstream segments by trapping sediments and reducing the sediment transport capacity because of the strong reduction of peak flows. The downstream geomorphic and ecological effects of dams are largely determined by the relative changes in the sediment transport regime, with consequences on channel incision and bank instability [32–34].

The detailed studies conducted by INAG (Portuguese water institute) [31] concerning the decision of whether or not to authorize the extraction activities allowed intense ecological impacts to be conclusively determined, demonstrating that the lowering of the estuary bed exceeded 7 m in depth in some points. Such studies also demonstrated that gravel mining, which ceased effectively in 1992, was environmentally unsustainable, and no further authorizations were processed, except for maintaining navigation conditions in the harbor. However, the morphological adjustments necessarily continued dramatically, which can also be explained by the fact that all 19 identified exploitation extractions exceeded the legal limits, driving estimated total values reaching 600,000 m$^3$/year [31]. In addition, the constant pressure on the marshes and on the banks of the lower segment of the river led to the loss of important habitats for conservation.

The survey carried out in situ and by aerial photography led us to consider that interventions with long-term purposes could not be limited to the consolidation of banks, but that they also should modify hydrodynamics because of the continued process of excavation of the lower layers of the bank. These display a complex structure, with a less cohesive layer at the base, affecting the stability of the entire bank. These aspects are similar, both in Cardielos (Figure 2, left) and Portuzelo (Figure 2, right), where the progressive erosion has led further to the collapse of the riparian layer.

This part of the estuary, downstream of the previous section, displays a higher vulnerable condition, which is due to the fact that the lowering of the estuarine bed caused by dredging reached a considerably higher depth close to the banks (the channel deepening reached here is 4–6 m).

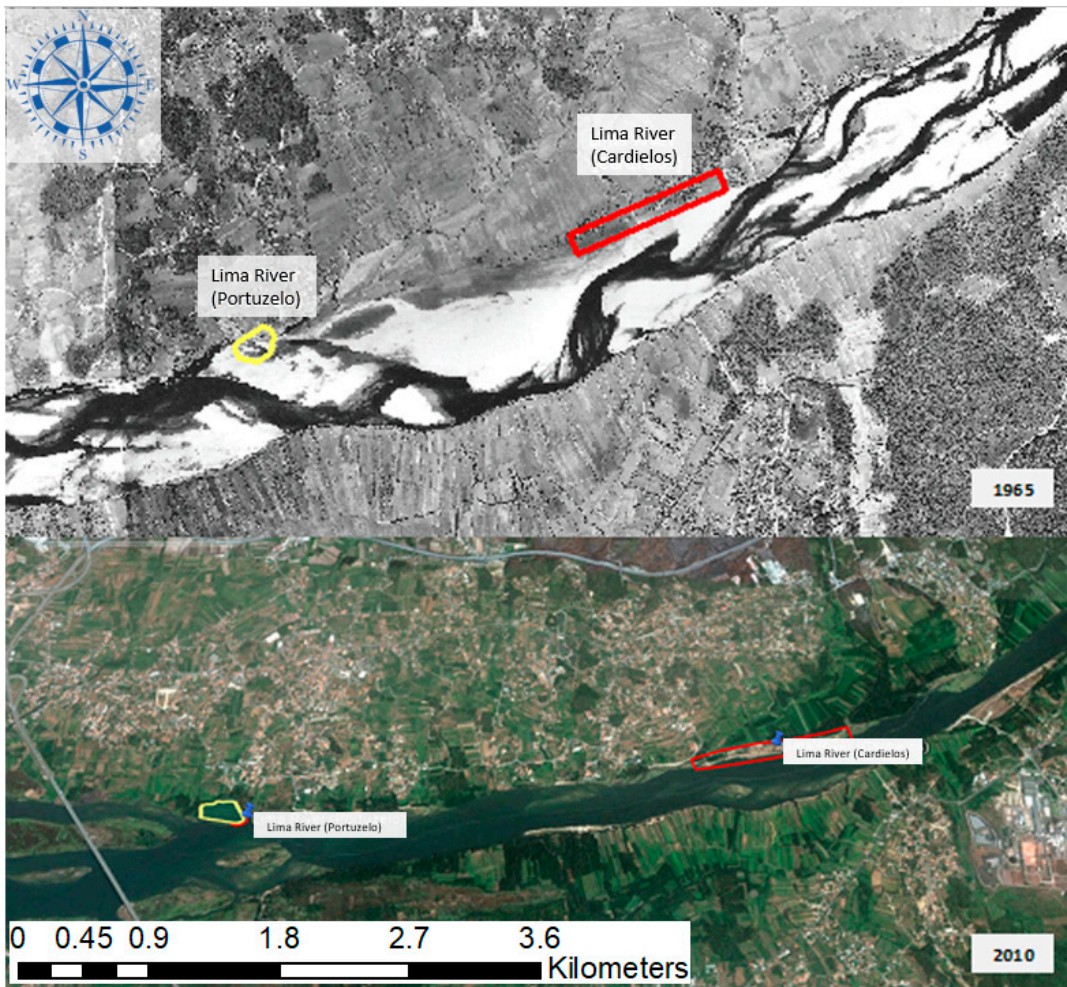

**Figure 1.** Aerial photographs of a part of the R. Lima estuary with the localization of the two considered priority segments for intervention: the comparison of the year 1965 with meander characteristics and the year 2010 without sediment in the river is shown. The yellow line intervention areas represent Portuzelo and red represents Cardielos.

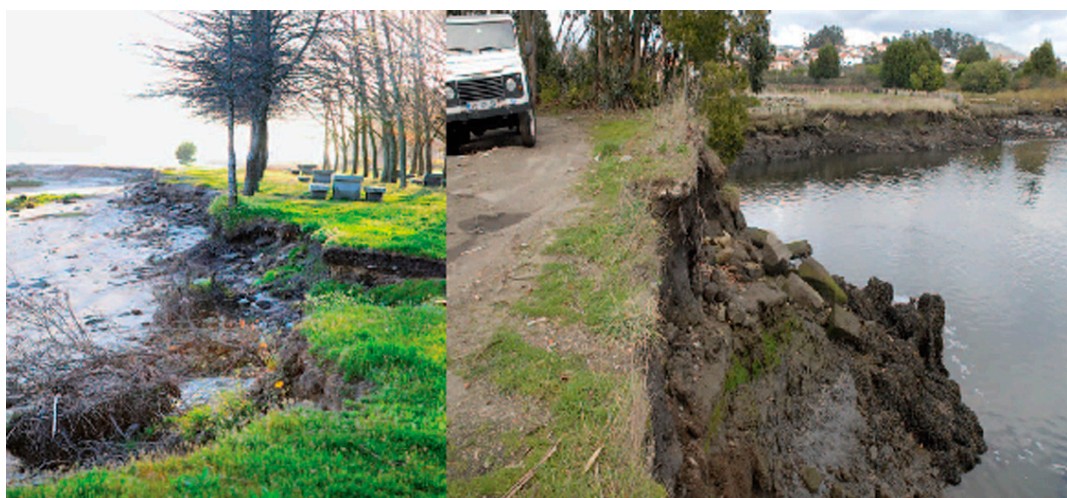

**Figure 2.** Cardielos (left panel) and Portuzelo (right panel) sections in 2010, previous to the bank reinforcement, where an apparent bank collapse is visible.

## 3. Results

There were multiple purposes of this project besides bank protection. We are aware that some of the civil engineering techniques used may have negative aesthetic implications, so there was concern about visual mitigation; besides, the rehabilitation should improve the riverine habitats, allowing a revegetation process towards a riparian gallery. Therefore, in this rehabilitation, we integrated different concepts of bank protection and hydrodynamics processes through the selection of convenient engineering techniques, with the additional purpose of stopping the irretrievable loss of an area of high biodiversity (wetlands/marshes) and marginal leisure infrastructures. The promotion of hydraulic roughness to progressively increase accretion (and, indirectly, salt marsh recovery) was inherent to the conception of the project.

### 3.1. Cardielos Section

The project was developed along with two temporal phases carried out between 2011 and 2013. It was defined as the first set of six triangular groynes, which were located in the most eroded segment. Afterwards, we built a second layer of groynes, closer to the water level, with a smaller dimension, which was further disposed along the same segment to complement and strengthen the first layer. This field of two lines of groynes, implanted along approximately 1 km, then formed a group of structures that acted together with the objective of causing the water to flow some distance from the riverbank and to increase hydraulic roughness. A groyne increases the roughness of the bank on which it is constructed and, in doing so, creates a zone of lower flow velocity in which the tendency for erosion is less and the deposition greater. Typically, eddy currents form in the pools between groynes where the water flows upstream along the bank [35]. These are wall-like structures, perpendicular to the flow direction and pointed towards the edges where the nose of the groyne is gently sloping.

Both sets of layers were built with rip-rap material, whereas the second line removes their visual impact since it is below the waterline at high tide. This group of structures (Figures 3 and 4) include granite rocks of 0.5–0.8 m in diameter packed in a layer thickness between 1.5–1.9 m and creates structures ranging in length from 13–29 m, depending on the topographical conditions where they are implanted. The second set (closer to the water) was composed of material with similar diameters, but packed around an axis of material with a small grain size (20–30 cm) and placed over a synthetic mat.

Because this set was placed in a plane which was more exposed to tidal and river flow energy, it was planned that the foundation of the structure should be place at a level close to the depth of the expected scour; this level was indicated by a careful observation along this section of the river. The defined layout (straight in plan and perpendicular orientations) allows the set of groynes to trap a moderate amount of sediment upstream and downstream, keeping the current more or less parallel to the bank and offering a medium potential for scour at the head. Both sets of groynes were rooted successively in a set of structures, which are listed in order from water level as follows: a) rip-rap between 3–5 m long; b) gabion mattress (placed on a gravel layer after shaping the bank), which is characterized by a wire basket filled with rock (covered with 20 cm of soil for planting and a wire mesh to decrease tidal washing); c) vegetation roll and willow fascine; and d) a gravel layer with soil (40 cm) covered by tridimensional geomats after bank reprofiling (Figure 4).

Finally, the described structures were vegetated with autochthonous hygrophytic and salinity-resistant herbaceous and woody species, such as reeds and rushes, combined with semi-halophyl macrophytes and salinity-resistant shrubs (*Juncus maritimus*, *J. acutus* or *J. effusus*, *Typha angustifolia*, *Phalaris arundinacea*, *Agrostis stolonifera*, *Scirpus maritimus*, *Festuca arundinácea*, *Phragmites* spp., *Tamaryx tamaryx*, *Carex* sp. or *Najas* spp.). In the vegetation roll and willow fascine layer, over the severely eroded area, we also conducted hydroseeding since the area is a space which is intensely used by visitors. Table 1 shows the techniques involved at the Cardielos site, including the floristic composition associated with the specified structures.

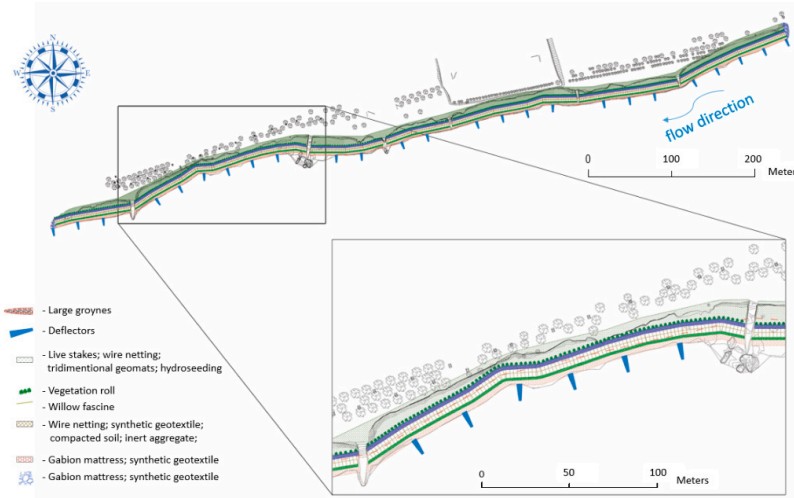

**Figure 3.** General view of the intervention area at the Cardielos site.

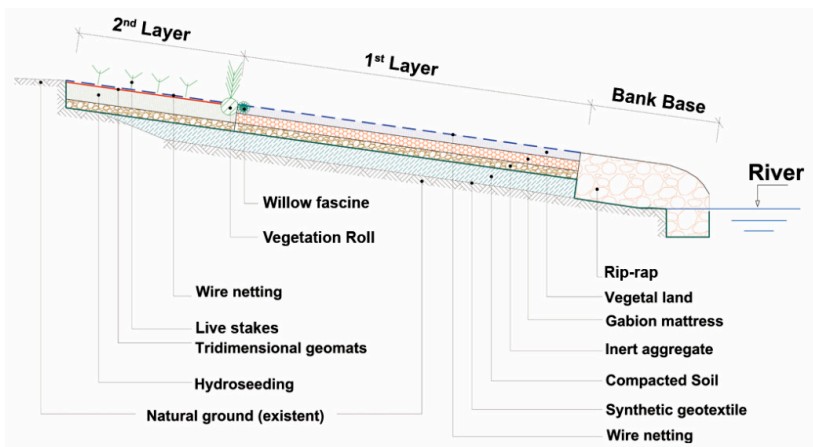

**Figure 4.** Planned techniques and vegetation species along the bank profile designed for the Cardielos site.

**Table 1.** Description of the planned techniques involved in each layer for the Cardielos section.

| Bank Profile | Techniques | Species |
|---|---|---|
| Bank base | Small groynes | - |
| | Rip-rap (50/80 cm) | - |
| | Large groynes | - |
| First layer (close to the water) | Live stakes | Salinity-resistant shrubs (*Juncus* spp., *Typha angustifólia, Phalaris arundinácea, Agrostis stolonifera, Scirpus maritimus, Festuca arundinácea, Phragmites* spp., *Tamaryx tamaryx*) |
| | Gabion mattress | - |
| Second layer | Willow fascine | *Salix atrocinerea* |
| | Vegetation roll | Semi-halophyl macrophytes |
| | Tridimensional geomats | *Salix atrocinerea; Salix salvifolia* |
| | Hydroseeding | *Lolium perenne; Festuca pratensis; Poa pratensis; Lolium multiflorum; Lupinus luteus; Dactylis glomerata; Trifolium subterraneum* |
| | Live stakes | *Juncus* spp., *Salix atrocinerea; Salix salvifolia* |

### *3.2. Portuzelo Section*

The techniques designed and implemented for this area, with lengths of approximately 150 m, are schematized in the profile shown in Figure 5 and Table 2. Again, the objectives, besides bank stabilization, allowed for the settlement of vegetation and increased the roughness on the submerged bank in order to trap sediments and to dissipate the energy from river flow and tidal dynamics, contributing to long-term sustainability. Besides, this bank constitutes a barrier that protects a large salt marsh. Therefore, it is a crucial aspect of this defense system for the preservation of this sensitive environment. As Figure 5 shows, from the base to the top of the bank different layers, we successively used a) rip-rap with large boulders (0.6–0.8 m) in a foundation frame of wood piles, with stakes driven into the riverbed (since the river depth was higher when compared to the previous section) in order to promote roughness (groynes were not considered because of the water depth); b) bank reprofiling to smooth the slope, which was covered with a layer of geogrids, filled in the lower part with gravel (for adequate infiltration) followed by soil and further vegetated, where reeds were placed near the base and woody vegetation (mainly *Tamaryx* sp.) was planted in the upper layer, and finally a wire mesh was used to decrease the potential tidal washing; and c) top lining and plantation with willow species as well as a row with broadleaf trees to improve the landscape attractively for visitors and to increase the overall bank consolidation. Besides this, we installed drains to allow the water to flow between the estuary and marshland.

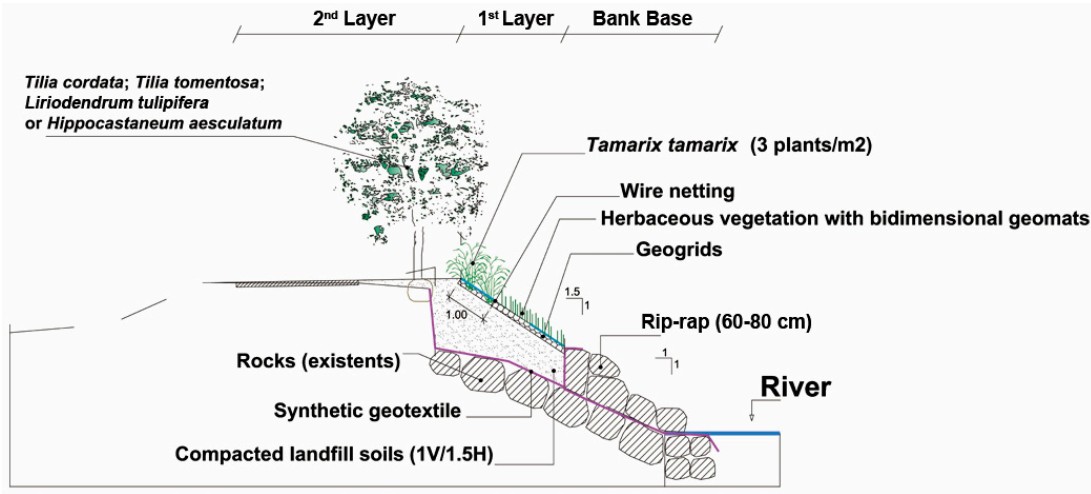

**Figure 5.** Planned techniques and vegetation species along the bank profile designed for the Portuzelo site.

**Table 2.** Planned techniques and vegetation species along the bank profile designed for the Portuzelo section.

| Bank Profile | Techniques | Vegetation Species |
|---|---|---|
| Bank base | Rip-rap (60/80 cm) | - |
| First layer (close to the water) | Geocells | - |
| | Bidimensional geomats (synthetic and organic) | - |
| | Live stakes | *Tamarix tamarix*; *Juncus* spp. |
| Second layer | Planting | *Salix atrocinerea*; *Salix salviifolia* |

Following this, a hydraulic study was conducted (using the HEC-RAS model) to compare the hydraulic conditions of the bank before and after the projected intervention, in order to estimate the energy dissipation of the flow energy. The hydraulic modelling enabled us to simulate various scenarios; in particular, the effect of increasing the hydraulic roughness to provide sedimentation

and the inherent stabilization of the bank toes. Consequently, these actions contributed indirectly to creating the conditions for the natural colonization and resettlement of marginal vegetation.

## 4. Discussion

This study includes local mitigation actions which have been applied to solve the most dramatic erosion problems in specific estuarine area sections. Of course, it would be more convenient to adopt a global management plan for the restoration of the entire estuary considering the stressing agents. However, the design presented here in the two considered areas represents a process to defend the salt marshes in this protected area, which we assume that will act as a motivation for extension to the different impacted areas of this estuary. Besides this, the inherent value of salt marshes for biodiversity, in the R. Lima estuary is that they also act as filters trapping or accumulating heavy metals, especially in the more densely vegetated high marsh layers [36], which play a significant role in dealing with the industrial effluents discharged upstream. More holistic approaches take advantage of conceptual models such as the one presented by Bergh's team in [37], where the dysfunctional patterns in habitat and community structures were traced back to anthropogenic changes in the physical and chemical processes, with the identification of key parameters and distinct rehabilitation proposals.

In these specific areas, we adopted soft engineering solutions to coastal flooding, namely by incorporating the planting of marsh vegetation in the intertidal zone for the purpose of promoting sedimentation and dissipating wave energy. We also followed the principles of Morris [38], for whom a successful design would employ plant species with varying degrees of tolerance to flooding, maximum drag, broad vertical ranges within the intertidal zone and which form a successional series. However, each rehabilitation method has to be observed under its specific conditions: if we provided the conditions for accretion because of a sediment deficit, other situations may require an inverse approach. This was the case for Garcia-Novo's team [39], which projected a hydraulic scheme favoring sand deposition upriver, avoiding its transfer to the Donãna marshes (South Spain) in order to prevent the excess of silting during flood events, which caused an unstable substrate with a lack of vegetation.

Thus, to estimate the hydraulic differences in order to analyze the ability to dissipate the energy created by the introduced structures, we computed the shear stress and current velocity for different recurrence periods, between the initial situation and considering the disposed of sets of groynes (Table 3).

**Table 3.** Values obtained by simulation with the HEC-RAS model to compare hydraulic parameters before and after the intervention.

| | | Before | | After | |
|---|---|---|---|---|---|
| **Manning Roughness (n)** | | **0.030–0.050** | | **0.023–0.036** | |
| **Cross-Section** | **Return Period (years)** | **Shear Stress (N/m$^2$)** | **Current Velocity (m/s)** | **Shear Stress (N/m$^2$)** | **Current Velocity (m/s)** |
| (Cardielos) | 2.33 | 6.13 | 0.50 | 4.34 | 0.58 |
| | 5 | 7.56 | 0.57 | 5.45 | 0.66 |
| | 100 | 14.58 | 0.84 | 10.76 | 0.98 |
| (Portuzelo) | 2.33 | 10.66 | 0.68 | 3.93 | 0.61 |
| | 5 | 14.05 | 0.79 | 5.19 | 0.72 |
| | 100 | 25.51 | 1.10 | 10.39 | 1.1 |

The hydraulic simulation was conducted to evaluate the results of flow magnitudes corresponding to two frequent events (2.33 and 5 years) and one extreme event (100 years). The hydraulic model developed adopted a range of manning values, n (Table 3), in order to calibrate the model based on the reference data and the conditions observed in situ.

We may conclude that there was a significative reduction of shear stress, which reached about 65%, corresponding also to the estimated lower current velocities, as a consequence of increasing the resistance to flow (displayed by Manning coefficients), which may also act as a sediment trap, protecting the base of the bank. However, as a result of the type of solutions implemented in the Cardielos area, there was an increase in speed; nevertheless, there was no risk of bank collapse.

Following the appropriate post-appraisal of the implemented project in the target areas of Cardielos and Portuzelo, we may draw some conclusions and recommendations. In the first 2 years after the project's conclusion, we could observe that, in Cardielos, all the structures showed a convenient resistance to critical environmental conditions. This is the case for the two rows of groynes, as well as the rip-rap or the gabion mattress (Figure 6), representing, therefore, a convenient solution since the erosion impact also decreased substantially and created the required barrier for bank protection preserving the built leisure structures. Besides this, the subsequent field surveys allowed us to observe that no more obvious scour holes were formed around the groyne layers. However, we also must accept that not much sediment deposition was observed between these structures, in contrast to our expectations, which retarded the natural re-vegetation process. The less successful results were observed in the layer affected by to the tidal movement, where we noticed a low success of woody vegetation development as the stake rooting was deficient, probably because of the small size of this biological material (less than 30 cm in length). Another cause was the lack of protection in relation to trampling (people and animals). In the case of the layer in the upper bank, other than the influence of the tides, we could observe better results, with higher plant survival and floristic diversity. With regard to the area of Portuzelo, the robustness of the rock base protection was evident, as well as the stability of the plateau following the installation of the geogrid wall. However, the planting success was only relative, such as the natural colonization by macrophytes or herbs, but the viability rate was more intense with the plantations of shrubs based on tamarisk. At low tide, it was possible to check for the proper functioning of the installed drains which were integrated into the created protection structure, which allowed the water to flow into the marshland, keeping a constant water level in this ecosystem, which essentially contributes to the sustainability of this wetland (Figure 7).

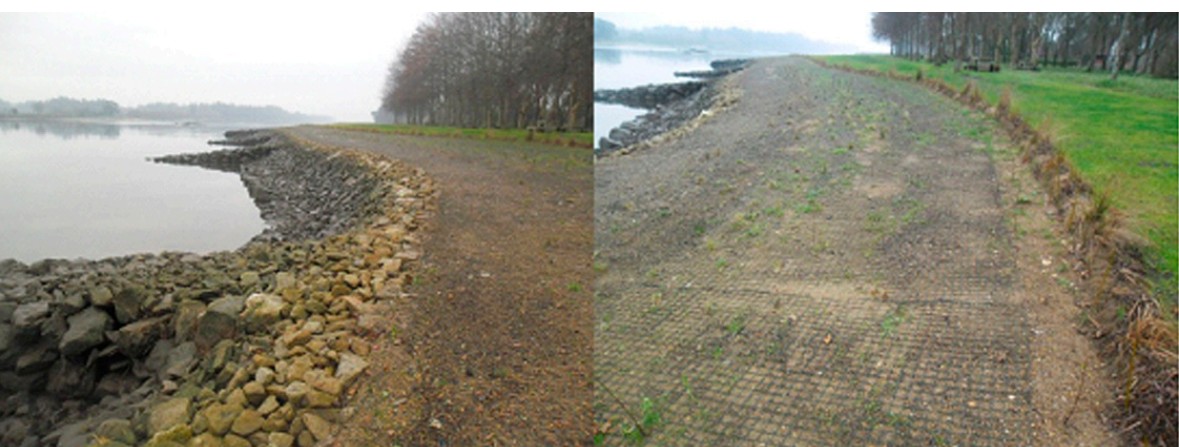

**Figure 6.** Rip-rap (left) and first row of small groynes (right) in Cardielos, after the implementation of the project in Cardielos in 2014.

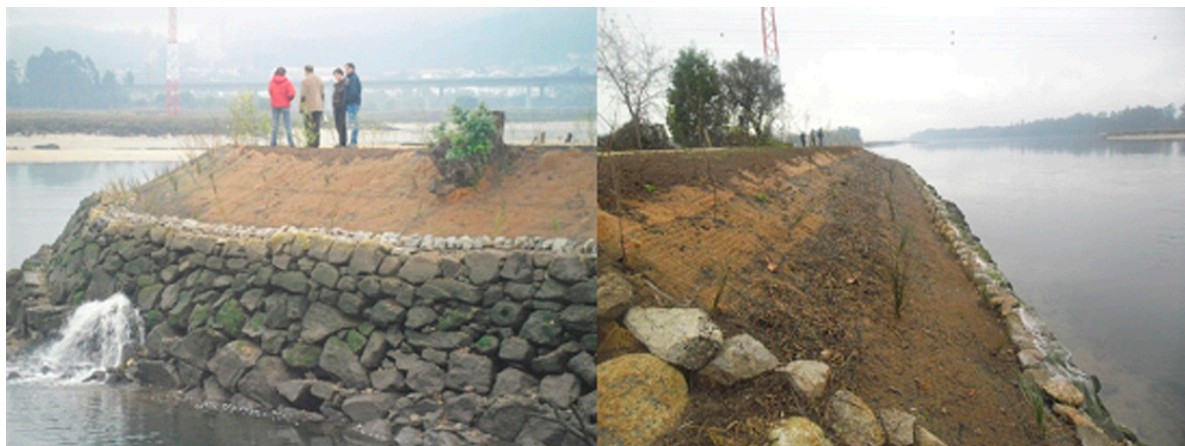

**Figure 7.** Rip-rap (left) and geogrid disposal (right) in the Portuzelo site (2014).

Of course, this action was focused in a specific part of an overall degraded estuarine environment. Immediately upstream and downstream of the rehabilitated sections, there is still a constant progression of the pressure on the banks and the consequent set-back of the bank line, which is reflected in Figure 8.

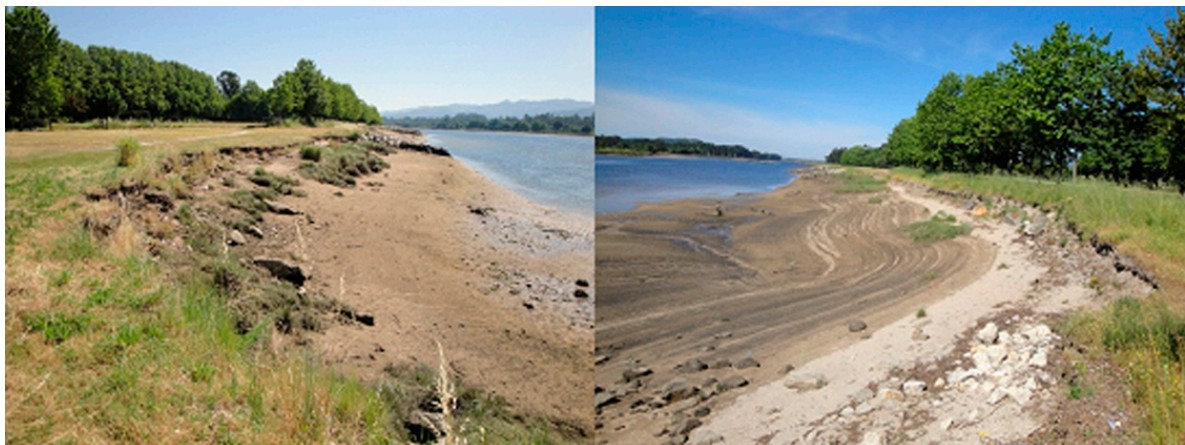

**Figure 8.** Two segments of the estuary banks, adjacent to the rehabilitation project in Cardielos (2014), showing that the severe erosion still progresses along the un-revetted banks, either in the upper section (left) or in the lower section (right), which requires an extension of the techniques already implemented.

Finally, we must stress that this is only a mitigation action, even if integrative; it will require a more complete study at a larger regional scale in the future, including proper actions in the entire estuary and even at the catchment level, in order to include the appropriate management actions that may contribute to overcoming the deficit of sediments in the estuarine area. For instance, Jacobs' workgroup [40], to restore tidal marshes on sites with low elevation, used a technique of controlled reduced tide (CRT), restricting the tidal regime with neap and spring tides by using high inlet culverts and low outlet valves, allowing the restoration of typical tidal freshwater vegetation. The choice between the advantages of intensive versus extensive ecological restoration should always be considered, and an interesting contribution to this subject was analyzed in terms of community biodiversity, successional changes, and costs by Gallego Fernandez and García Novo [41] in the restoration of a tidal marsh in SW Spain. Here, they compared high and low-intensity interventions, allowing us to conclude that the extra cost of building heterogeneous habitats in the intense intervention bore no relation to results. Ecological engineering is a very promising approach to maintaining intertidal marshes in equilibrium, but we believe that it has to incorporate a larger area of the estuary. This is also the concept of Danielsen's group [5] and Morris [38], who adopted the extensive planting of marsh vegetation in the intertidal

zone to promote sedimentation and dissipate wave and river energy, with the additional positive benefits of accreting sediment.

Much research has focused on the importance of vegetation floodplains to create transient storage for channel sediments, becoming efficient traps—also for pollutants—and avoiding streambank retreat [42,43] (see Curran and Hession [44] for a compilation of the vegetative impacts on hydraulics and sediments). The sediment trapping ability of the vegetation allows for more growth and consequently further deposition; in [45], it was observed that plant succession could lead even to softwood forest establishment.

We must point out that if gravel extraction were to seriously impact the upper part of the estuary, reducing the salt marsh area—an activity that is now forbidden, in the most downstream part—a 3 km navigational channel would be maintained by regular dredging activities, which now will also cause the destruction of the wetlands and changes in sediment composition (the enrichment of fine sediments with high organic matter content), where typical floristic communities have been washed away [46]. We intend that the partial rehabilitation techniques presented here may constitute a stimulus to a more global management action aimed at the protection of salt marshes in this important hot-spot; however, more consistent restoration requires another scale and the coordination of different river authorities. This is indeed a critical overview of this project. For instance, the dredged material from the lower estuary (the mentioned channel for navigation) could be moved into the eroded river bank to mitigate incision, where the built groynes and deflectors (particularly at Cardielos) could promote the sedimentation and stability of the inserted gravel. The monitoring of this bedload transport, namely by particle tracking via radio telemetry [47], could allow us to obtain transport paths and increase the efficiency of this procedure.

## 5. Final Remarks

Finally, we share the opinion of González del Tanágo's team [32], arguing for a more holistic approach to water resources and land-use management at the catchment scale in order to understand the synergistic effects of dams, sediment supply and vegetation growth to implement the appropriate management and rehabilitation actions. The authors stress very different processes and geomorphic consequences in Iberian rivers, namely gravel-bed systems as in the case of the Lima river, in which sediment deficit downstream of the dams has triggered channel incision, and other Mediterranean streams where river regulation, in contrast, resulted in channel narrowing. Here, long-term photographic registrations allowed us to conclude that there was an increase of aggradation processes and vegetation encroachment, because of the reduction of the geomorphic discharges which are able to transport fine sediment downstream from the dams and the high sediment delivery of the catchment promoted by agricultural development. These aspects, finally, show the necessity of adopting specific rehabilitation processes adapted to each catchment, according to soil use patterns, flow changes and geological and physiographic features and the important of avoiding generic solutions.

**Author Contributions:** Conceptualization, L.F.S.F., A.A.S.P. and R.M.V.C.; Methodology, A.A.S.P.; L.F.S.F. and R.M.V.C.; Validation, L.F.S.F. and R.M.V.C.; Supervision, L.F.S.F. and R.M.V.C.; Project administration, R.M.V.C.; Funding acquisition L.F.S.F., F.A.L.P. and R.M.V.C.; Software implementation, A.A.S.P. Data curation and processing, A.A.S.P. and D.P.S.T.; Writing—original draft preparation, A.A.S.P.; Writing—review and editing, D.P.S.T. and F.A.L.P. All authors have read and agreed to the published version of the manuscript.

**Funding:** For authors integrated with the CITAB Research Centre, this work was further financed by the FEDER/COMPETE/POCI—Operational Competitiveness and Internationalization Program, under Project POCI-01-0145-FEDER-006958, and by the National Funds of FCT—Portuguese Foundation for Science and Technology, under the project UID/AGR/04033/2019. For the author integrated in the CQVR, the research was additionally supported by the National Funds of FCT—Portuguese Foundation for Science and Technology, under the project UID/QUI/00616/2019.

**Acknowledgments:** The authors are grateful to the Viana do Castelo Council who supported the project's conception and implementation and the office "Formas & Conceitos" who collaborated in all the steps of the project appraisal, from baseline surveys to option evaluation and technical design and project documentation preparation.

**Conflicts of Interest:** The authors declare no conflict of interest. The funders had no role in the design of the study; in the collection, analyses, or interpretation of data; in the writing of the manuscript, or in the decision to publish the results.

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
