# Peer review of "Combination of Ecological Engineering Procedures Applied to Morphological Stabilization of Estuarine Banks after Dredging"

_water, doi:10.3390/w12020391_

Round 1

Reviewer 1 Report

Review of “Combination of ecological engineering procedures applied to morphological stabilization of estuarine banks after dredging”

This paper provides a case study of bank stabilization on an actively eroding estuary in Portugal. Streambanks have become unstable due primarily to dredging/gravel extraction, which has deepened the channel and increased bank heights. Additionally, two upstream dams are likely trapping sediment and preventing it from replenishing the streambed. Stabilization at two sites consisted of a combination of hard and soft engineering techniques. Rip-rap, groynes, and gabions were used as structural controls, while a variety of vegetation was planted to increase stability, increase flow roughness, and induce deposition. Results generally show that the banks were stabilized, although there is less sediment deposition than expected and some vegetation plantings were unsuccessful.

General comments:

While the assessment of ecological engineering approaches for stabilizing eroding banks in estuaries is interesting, there are several major issues with the paper that need to be addressed.

First, the Introduction doesn’t clearly outline the motivations for this study. In Line 58, the authors state that “The purpose of this study is to analyze the erosion phenomena…”. However, there is very little analysis of the erosion that is occurring, and the paper mostly focuses on describing two small-scale bank stabilization projects. It isn’t clear if these are new techniques that have not yet been applied in this area, and if the goal is to test these techniques to see if they are suitable for application on other banks in the Lima River estuary. Furthermore, much of the Introduction focuses on the importance of salt marsh habitat, and implies that a major goal of this project is protection of this habitat. While this may be true at the Portuzelo site where there is a salt marsh, this does not appear to be the case at the much larger Cardielos site where no salt marsh is present. If the goal was to create salt marsh at this site, this is not clear throughout the paper. Protecting riparian forest habitat is also important, but currently too much of the focus is on salt marshes when they don’t seem to be that large of a part of the project.

The paper is also not well organized. The Results section is really more appropriate for Methods. The description of the stabilization projects, the monitoring methods, and HEC-RAS analysis are all Methods, not Results.

With this change, there aren’t really many substantive results in this paper. For example, many of the “Results” are just visual observations following project completion. However, there are no quantitative measurements (or even estimates) of project performance. Was there any pre- and post- project surveying to quantify rates of bank erosion and/or sedimentation? What about vegetation survival rates? Was any structured monitoring done at regular intervals or were visual observations just made 2-years following project completion? Qualitative observations are useful, but should be backed by some evidence. The main quantitative results of the paper are the HEC-RAS modeling outputs (which should be in the Results section, not the Discussion).

These HEC-RAS results are useful, but there is no detail on the model approach. There needs to be much more information on these methods including the spatial extent of the model, what data were used, and what conditions where different between the “Before” and “After” modeling (primarily Manning’s n values, I assume). How were these roughness values selected? Why were the three return periods chosen, how were these flows calculated, and what were the values? Was sediment transport actually modelled or did you just infer changes in sediment transport/deposition based on calculated shear stress values? Were these shear stresses cross section averages or at the banks? Furthermore, there needs to be more of a description of why, if you calculated these reductions in shear stress with bank roughening, did you not observe sedimentation? How do these shear stresses compare to the critical shear stress for mobilization of the sediment sizes you hoped to trap? Also, why weren’t lower flow magnitudes analyzed. Looking at large events is useful for determining whether additional scour might happen, but typically deposition would be most likely during smaller flow events. Furthermore, on page 9, lines 237-240, you state that both shear stress and velocity decreased after restoration. However, velocities at the Cardielos site actually increased. Why do you think this is and what implications does this have for sediment trapping?

The Discussion doesn’t seem appropriate for the observations made. Much of the Discussion focuses on studies on salt marsh restoration, but it still isn’t clear that this was a main goal of this project, or that monitoring was conducted specific to salt marsh establishment or protection. The authors conclude by discussing the need for catchment-scale approaches to restore the Lima River estuary; however, they provide very few specifics on what this would look like for this system (e.g. what main issues need to be addressed, and how would they address them).  

Another significant concern is that the paper topic doesn’t seem to fit with the goals of the special issue on “Water Security and Governance in Catchments”. The description of the special issue focuses mainly on water security including water shortages, flooding, and human dimensions. It isn’t clear how bank stabilization in estuaries (and potentially salt marsh protection/restoration) relates to this topic. There is the brief mention of “ecosystem services” in the special issue description, but there is still not a clear link between the motivations of this paper and the special issue.

Finally, the writing in many places is unclear, caused mainly by grammar and language issues. I have tried to note the areas most in need of revision in the attached pdf.

Additional Comments:

Page 2, Lines 52-53: The authors state that ecological engineering has rarely been applied in estuarine areas. However, there is a long history of ecological engineering associated with salt marshes and coastal protection (including papers that the authors cite elsewhere in the paper). Is the point of this statement that projects specifically to stabilize estuarine banks are not common?

Page 2, Lines 52-53: Many of the citations used to support the statement that ecological engineering has been used to stabilize river banks [8-12] are related to bank erosion modeling and processes, and don’t actually discuss stabilization and ecological engineering. There is a wide body of knowledge on bioengineering approaches to bank stabilization that is not cited here.

I’ve included additional comments throughout the pdf (attached).

Author Response

Reviewer #1:

General comments:

While the assessment of ecological engineering approaches for stabilizing eroding banks in estuaries is interesting, there are several major issues with the paper that need to be addressed.

1- First, the Introduction doesn’t clearly outline the motivations for this study. In Line 58, the authors state that “The purpose of this study is to analyze the erosion phenomena…”. However, there is very little analysis of the erosion that is occurring, and the paper mostly focuses on describing two small-scale bank stabilization projects. It isn’t clear if these are new techniques that have not yet been applied in this area, and if the goal is to test these techniques to see if they are suitable for application on other banks in the Lima River estuary. Furthermore, much of the Introduction focuses on the importance of salt marsh habitat, and implies that a major goal of this project is protection of this habitat. While this may be true at the Portuzelo site where there is a salt marsh, this does not appear to be the case at the much larger Cardielos site where no salt marsh is present. If the goal was to create salt marsh at this site, this is not clear throughout the paper. Protecting riparian forest habitat is also important, but currently too much of the focus is on salt marshes when they don’t seem to be that large of a part of the project.

The objectives were clarified. The main objective it is not to analyse erosion dynamics but to create the conditions to promote the re-establishment of the lost salt marshes and to protect the existing ones. The reviewer is wright in pointing out the some difficulty in assessing the targets of this work. Like it is more detailed now, this is a Nature 2000 area (devoted to protect the riverine habitats), but influenced by different sources of disturbance, where the most significant conservation values are provided by the salt-marshes. The remnant ones of these habitats should be protected against erosion and, at the same time, we want to induce the system in accommodating salt marshes and wetlands where they previously existed. A reference was added (Costa-Dias et al., 2010) to support our statements.

2-The paper is also not well organized. The Results section is really more appropriate for Methods. The description of the stabilization projects, the monitoring methods, and HEC-RAS analysis are all Methods, not Results.

This paper aims to present the results of the implementation of a river rehabilitation project and, in this sense, the main technical solutions applied, including the corresponding technical specifications, were described in the Results chapter, based on the reference conditions of the site intervention. However, in order to make the article more organized, some changes were made to the text, allocating some text from the Results to the Methods chapter. The analysis of the results is carried out in the discussion chapter, based on the field observations made (monitoring) and the application of the Hec-Ras program by comparing the results obtained before and after the intervention.

With this change, there aren’t really many substantive results in this paper. For example, many of the “Results” are just visual observations following project completion. However, there are no quantitative measurements (or even estimates) of project performance.

3-Was there any pre- and post- project surveying to quantify rates of bank erosion and/or sedimentation?

As part of this work, no survey or quantification of the erosion / sedimentation rate was carried out. Only an assessment methodology was applied through visual observation in situ. However, it is intended, over time, to carry out work in this guideline, as referred to in the article (lines 293-296 of the new manuscript).

4-What about vegetation survival rates?

As mentioned in the previous comment, there was no quantitative analysis of the results obtained, beyond the hydraulic evaluation through the application of the Hec-Ras program. However, it was possible to verify a high success rate in the vegetation species applied in bioengineering solutions (Table 1).

5-Was any structured monitoring done at regular intervals or were visual observations just made 2-years following project completion?

This case study was subject to regular monitoring after intervention, either by the project team, or as part of a doctoral thesis (Pinto, 2018), and the methodology for monitoring riverbank stabilization interventions was presented in the referenced article (Pinto et al, 2016).

6-Qualitative observations are useful, but should be backed by some evidence. The main quantitative results of the paper are the HEC-RAS modeling outputs (which should be in the Results section, not the Discussion).

The authors admit that the organization of the paper could have been different. However, in the results chapter, it was decided to present the rehabilitation project implemented and in the discussion chapter to present an analysis of the evaluation of post-intervention results, based on the application of the HEC-RAS program.

7-These HEC-RAS results are useful, but there is no detail on the model approach. There needs to be much more information on these methods including the spatial extent of the model, what data were used, and what conditions where different between the “Before” and “After” modeling (primarily Manning’s n values, I assume).

Having a limited number of characters for the elaboration of the paper, we tried to present only the fundamental information for the explanation of the methods used in the evaluation of the results. As mentioned in lines 258-261(of the new manuscript), the hydraulic simulation was based on the variation of the roughness values (Maninng) corresponding to the situations before and after the intervention (reference values given by the Hec-Ras user manual adjusted to the situation under study) . For each case study, the total extent of the intervention was modeled.

8-How were these roughness values selected?

The roughness values used were selected by the user manual of the Hec-Ras program adjusted to the situation under study.

9-Why were the three return periods chosen, how were these flows calculated, and what were the values?

The choice of return periods was based on the frequency of occurrence of the flow corresponding to the first two selected (2.33 and 5 years) and on the assessment of the behavior of an extreme event (100 years). The flow rates used were calculated based on equations defined in the Hydrographic Region Management Plan (PGRH-RH1) adjusted to the Lima River watershed.

10-Was sediment transport actually modelled or did you just infer changes in sediment transport/deposition based on calculated shear stress values?

Sediment transport modeling was not performed. The analysis was based only on the evaluation of the results of the hydraulic simulation and visual observations in situ.

11-Were these shear stresses cross section averages or at the banks?

The values shown correspond to the drag tension at the margin.

12-Furthermore, there needs to be more of a description of why, if you calculated these reductions in shear stress with bank roughening, did you not observe sedimentation?

As mentioned in lines 269-271 (of the new manuscript), the deposition of sediment was verified, however, more reduced than initially expected. This the situation may have happened, since this case study is located in an estuarine area with the effect of the seas feeling on the banks.

13-How do these shear stresses compare to the critical shear stress for mobilization of the sediment sizes you hoped to trap?

We believe that by decreasing bed shear stress we avoid the scouring of coarse material and at the same time we promote the sediment entrainment by the structures created

14-Also, why weren’t lower flow magnitudes analyzed. Looking at large events is useful for determining whether additional scour might happen, but typically deposition would be most likely during smaller flow events.

As mentioned in the reply to comment 9, the flow rates corresponding to the most frequent return periods in the hydrographic basin (2.33 and 5 years) were analyzed, in order to take this into account.

15-Furthermore, on page 9, lines 237-240, you state that both shear stress and velocity decreased after restoration. However, velocities at the Cardielos site actually increased. Why do you think this is and what implications does this have for sediment trapping?

Shear stress is lower in both Cardielos and Portuzelo after the intervention and this is the critical parameter related to bank stabilization.

16-The Discussion doesn’t seem appropriate for the observations made. Much of the Discussion focuses on studies on salt marsh restoration, but it still isn’t clear that this was a main goal of this project, or that monitoring was conducted specific to salt marsh establishment or protection. The authors conclude by discussing the need for catchment-scale approaches to restore the Lima River estuary; however, they provide very few specifics on what this would look like for this system (e.g. what main issues need to be addressed, and how would they address them).

This aspect is related to the first one in the Introduction concerning the main goals of the project, which were better clarified in the introduction. Nevertheless, in the beginning of the Discussion we included some additional comments to stress that the preservation or recreation of the salt marshes represent the main objective of the techniques applied. A paper of Cardoso et al (2008) was included now and it refers specifically to the role of salt marshes in the R. Lima estuary.

17-Another significant concern is that the paper topic doesn’t seem to fit with the goals of the special issue on “Water Security and Governance in Catchments”. The description of the special issue focuses mainly on water security including water shortages, flooding, and human dimensions. It isn’t clear how bank stabilization in estuaries (and potentially salt marsh protection/restoration) relates to this topic. There is the brief mention of “ecosystem services” in the special issue description, but there is still not a clear link between the motivations of this paper and the special issue.

Regarding the choice of the special issue, we really could have chosen another one. But in reality, this seemed appropriate when submitting. Either way, we are open for it to be changed to the special issue that they think is most appropriate.

Finally, the writing in many places is unclear, caused mainly by grammar and language issues. I have tried to note the areas most in need of revision in the attached pdf.

Additional Comments:

18-Page 2, Lines 52-53: The authors state that ecological engineering has rarely been applied in estuarine areas. However, there is a long history of ecological engineering associated with salt marshes and coastal protection (including papers that the authors cite elsewhere in the paper). Is the point of this statement that projects specifically to stabilize estuarine banks are not common?

Page 2, Lines 52-53: Many of the citations used to support the statement that ecological engineering has been used to stabilize river banks [8-12] are related to bank erosion modeling and processes, and don’t actually discuss stabilization and ecological engineering. There is a wide body of knowledge on bioengineering approaches to bank stabilization that is not cited here.

In fact these nature-solution actions are uncommon in Portugal, especially in estuarine areas where the stabilization procedures are typical heavy artificialization ones, where no room is left for improving habitats. So we clarified this aspect.

I’ve included additional comments throughout the pdf (attached):

19-Line 30: It is unclear what you mean by this sentence. What is the "limit of the tides influence"?

We changed the expression

20-Line 40: It would be clearer if this sentence was split into two.

It has been corrected.

21-Line 50: It is unclear what you mean by this. Please clarify.

It has been corrected.

22-Line 61: What attempts have been made? Why have they been fruitless and why is your approach more likely to succeed?

The mention to the fruitless of the actions in this area were now excluded. In fact there was not till now a serious management plan. Our efforts (this project) were supported by the local administration in order to apply for a serious involvement of regional and national authorities.

23-Line 73: what are these units?

The units “hm” are hectometre.

24-Line 74: channel slope?

Considering average channel slope.

25-Line 76: This is a long and confusing sentence that could be split into two and clarified.

The sentence was split and improved.

26-Line 91: should this be "thalweg"?

The word "talveg" has been replaced by "thalweg".

27-Caption Figure 1: Could you provide more details on this figure? What are the differences in the yellow versus red lines? Is the yello at Portuzelo the marsh area that is being protected? Also, the figure needs a scale bar and north arrow.

Thank you for the observations in Figure 1, all the suggested details have been changed.

28-Line 109: This sentence is very unclear.

The in-depth studies carried out by INAG (water institute) regarding the sustainability of the pursuit of inert extraction activities in the Hydrographic Basins of the Lima and Cávado rivers, have shown that in the Lima river, in an appreciable part of the section between the Lanheses Bridge and the foz there is a situation of deep erosion, reaching in some cases a lowering of the bed that exceeds 7m.

29-Line 115: It is unclear what this means. What are the "exploitations extractions" and is 600,000 m3/yr the total amount of gravel extracted? Is that current? Above you say gravel mining stopped in 1992.

The extraction of aggregates stopped in 1992. Until 1992, the extraction of aggregates was approximately 600,000 m3 / year, which caused an enormous river dynamics in this water line.

30-Line 116: Is this erosive "pressure" or are you talking about something else?

Is this erosive “pressure”.

31-Line 120: Does this mean you have data on historic bank erosion rates? This would be important information to include in the paper.

We have data that show the variation of the longitudinal profile of the lower sector of the River Lima and that are cited (reference 24).

32-Line 133: Do you mean a "gallery forest"?

It has been corrected.

33-Caption Figure 3: To be helpful, this figure should include labels describing what different parts of the schematic represent. Be sure to include what is existing infrastructure/vegetation, and what was added as part of the intervention. Also, include a scale bar and arrow indicating flow direction.

Thank you for the observations in Figure 3, all the suggested details have been added. The entire area on this illustration has been intervened “(infrastructure/vegetation)”.

34-Line 161: What does this mean? You estimated maximum scour depth based on scour holes you observed elsewhere? What was this scour depth?

The depth to which the margin stabilization solutions were implemented was determined by observing in situ the different heights of water imposed by the effect of the tides.

35-Line 166: the area with rip-rap was 3-5 long, correct? This isn't the size of the stone? This could be clarified.

The average diameter (d50) of the stones to be used in the technical groynes solution is 50-80 centimetre. The location of the application of the groynes on the margin is in a range of 3 to 5 meters, as shown in figure 4 (bank base).

36-Line 200: This deserves more explanation. Is the marsh currently inundated by high tide? What purpose does the drain have? It seems like it would primarily have one-way flow (from the marsh to the estuary). Also, the drain doesn't allow sediment to access the marsh which is important for marsh maintenance.

The installed drains are designed to have two-way flow in order to drain the excess of water in flooding periods, corresponding to peak flows of R. Lima, but also to allow water input into the marshes during high tide keeping a relatively constant Water level. There is not a sediment deficit in the existing marshes in Portuzelo site, therefore this is not a target, on the contrary to Cardielos reach.

37-Line 214: This is a very vague sentence and it is unclear what you are saying.

The sentence was rephrased

38-Line 243: What does this mean? That the structures stabilized the bank?

Yes, the structures allowed a reduction of shear stress driving a sedimentation of fines (e.g. silt bars). Such stability allowed a natural colonization by vegetation, which increased the protection of the previous undercut banks. The objective was to change the hydrodynamic conditions in order that the system could achieve a dissipation of energy in order to create the conditions to stop the erosion and to accelerate colonization of vegetation, which could enhance the stability.

39-Line 260: Again, the role of the drain needs to be explained further.

We agree. This sentence was rephrased for a better comprehension of the role of the drains.

40-Caption Figure 7: Isn't this the Portuzelo section? That is how it is referred to in the text.

It was  a mistake. It has been corrected.

41-Caption Figure 8: I think you figure captions are off. Based on the text, these images are showing erosion of banks outside the two project sites.

It was a mistake. It has been corrected.

42-Line 292: OK, but how does this research relate to what you found (i.e. no sediment trapping). What does this previous work suggest about why your project was successful in this area?

We defend low-intensity interventions, this is the placement of structures to enhance rearing habitats and sedimentation to protect the river banks and to allow the resettlement of salt-marshes, instead of an excessive artificialization of all system. We believe that the references mentioned in the text support our concept.

43-Line 305: This sentence is unclear. Are you referring to the work of Gonzalez del Tanago?

The reviewer is right and this aspect was clarified.

Reviewer 2 Report

The authors present a practical case study around the use of green restoration approaches for the stabilisation of banks of estuaries after dredging. This work can be of interest to the broad readership of the journal but it is very important that the authors:

a) aim to generalise their findings on how these could apply to a broader range of environments or if any of the limitations/special aspects of their case study may increase the uncertainty in deriving useful results for other cases.

b) offer the manuscript for proofreading by a professional service - there are many grammar and English mistakes that may at times distract from clearly presenting their work.

Specific comments:

Line 22: Rephrase: “The consequences were the progressive impact along several decades of this protected area, even if the extraction activities ceased, with consequences on the preservation of salt marshes.”

Line 28: change “recover” to “recovery”

Line 29: grammar “did not allowed” to “did not allow”

Line 31: unclear what is meant with “until the limit of the tides influence. ” Rephrase.

Line 32: remove “a” from “a proper”

Line 44: “decreasing the impact of turbulence“ add “which can be important in entraining sediment, as shown in [Valyrakis et al. 2010; 2013”.

Valyrakis, M. , Diplas, P., Dancey, C.L., Greer, K. and Celik, A.O. (2010) Role of instantaneous force magnitude and duration on particle entrainment. Journal of Geophysical Research: Earth Surface, 115(F02006), 18p. (doi:10.1029/2008JF001247)

Valyrakis, M. , Diplas, P. and Dancey, C.L. (2013) Entrainment of coarse particles in turbulent flows: an energy approach. Journal of Geophysical Research: Earth Surface, 118(1), pp. 42-53. (doi:10.1029/2012JF002354)

Line 52: “… stabilize river banks” add the referenced below:

Yagci, O., Celik, M. F., Kitsikoudis, V., Ozgur Kirca, V.S., Hodoglu, C., Valyrakis, M. , Duran, Z. and Kaya, S. (2016) Scour patterns around isolated vegetation elements. Advances in Water Resources, 97, pp. 251-265. (doi:10.1016/j.advwatres.2016.10.002)

Liu, D., Valyrakis, M. and Williams, R. (2017) Flow hydrodynamics across open channel flows with riparian zones: implications for riverbank stability. Water, 9(9), 720. (doi:10.3390/w9090720)

Line 55: add the following references:

Redelstein, R., Zotz, G. and Balke, T. (2018) Seedling stability in waterlogged sediments: an experiment with saltmarsh plants. Marine Ecology Progress Series, 590, pp. 95-108. (doi:10.3354/meps12463)

Cao, H., Zhu, Z., Balke, T. , Zhang, L. and Bouma, T. J. (2018) Effects of sediment disturbance regimes on Spartina seedling establishment: implications for salt marsh creation and restoration. Limnology and Oceanography, 63(2), pp. 647-659. (doi:10.1002/lno.10657)

Line 101: reword: “significant higher energetic power. “

Text is not clearly legible in Figure 1. Enlarge text font and contrast.

Line 250: avoid using terms like “disappointing “. These are rather descriptive and subjective. Aim to use metrics to quantify how bad or good the conditions are where possible.

Author Response

Reviewer #2:

The authors present a practical case study around the use of green restoration approaches for the stabilisation of banks of estuaries after dredging. This work can be of interest to the broad readership of the journal but it is very important that the authors:

a) aim to generalise their findings on how these could apply to a broader range of environments or if any of the limitations/special aspects of their case study may increase the uncertainty in deriving useful results for other cases.

The main goals of the project were better clarified in the introduction.

b) offer the manuscript for proofreading by a professional service - there are many grammar and English mistakes that may at times distract from clearly presenting their work.

The authors agree and have carried out a more detailed review of English.

Specific comments:

Line 22: Rephrase: “The consequences were the progressive impact along several decades of this protected area, even if the extraction activities ceased, with consequences on the preservation of salt marshes.”

The phrase, was replaced in the text to “Leading to some consequences, such as the progressive negative impact on the preservation of salt marshes over several decades of this protected area, which continued even after the cessation of extraction activities.”.

1-Line 28: change “recover” to “recovery”

It has already been rectified.

2-Line 29: grammar “did not allowed” to “did not allow”

It has been corrected

3-Line 31: unclear what is meant with “until the limit of the tides influence. ” Rephrase.

The sentence has been reworded to: “The colonization of species (plants) in brackish and tidal water was a difficulty posed to this project”.

4-Line 32: remove “a” from “a proper”

It has been corrected

5-Line 44: “decreasing the impact of turbulence“ add “which can be important in entraining sediment, as shown in [Valyrakis et al. 2010; 2013”.

Valyrakis, M. , Diplas, P., Dancey, C.L., Greer, K. and Celik, A.O. (2010) Role of instantaneous force magnitude and duration on particle entrainment. Journal of Geophysical Research: Earth Surface, 115(F02006), 18p. (doi:10.1029/2008JF001247)

Valyrakis, M. , Diplas, P. and Dancey, C.L. (2013) Entrainment of coarse particles in turbulent flows: an energy approach. Journal of Geophysical Research: Earth Surface, 118(1), pp. 42-53. (doi:10.1029/2012JF002354)

The suggested references have been duly added.

6-Line 52: “… stabilize river banks” add the referenced below:

Yagci, O., Celik, M. F., Kitsikoudis, V., Ozgur Kirca, V.S., Hodoglu, C., Valyrakis, M. , Duran, Z. and Kaya, S. (2016) Scour patterns around isolated vegetation elements. Advances in Water Resources, 97, pp. 251-265. (doi:10.1016/j.advwatres.2016.10.002)

Liu, D., Valyrakis, M. and Williams, R. (2017) Flow hydrodynamics across open channel flows with riparian zones: implications for riverbank stability. Water, 9(9), 720. (doi:10.3390/w9090720)

The suggested references have been duly added.

7-Line 55: add the following references:

Redelstein, R., Zotz, G. and Balke, T. (2018) Seedling stability in waterlogged sediments: an experiment with saltmarsh plants. Marine Ecology Progress Series, 590, pp. 95-108. (doi:10.3354/meps12463)

Cao, H., Zhu, Z., Balke, T. , Zhang, L. and Bouma, T. J. (2018) Effects of sediment disturbance regimes on Spartina seedling establishment: implications for salt marsh creation and restoration. Limnology and Oceanography, 63(2), pp. 647-659. (doi:10.1002/lno.10657)

The suggested references have been properly added.

8-Line 101: reword: “significant higher energetic power. “

The sentence has been reworded to: “significant higher stream power”.

9-Text is not clearly legible in Figure 1. Enlarge text font and contrast.

Figure 1 quality has been improved.

10-Line 250: avoid using terms like “disappointing “. These are rather descriptive and subjective. Aim to use metrics to quantify how bad or good the conditions are where possible.

The sentence was properly reworded.

Reviewer 3 Report

Congratulations to authors. Very interesting paper regarding a current theme: an ecological way to minimize some consequences of gravel extraction in rivers.

In general the paper is well presented but some information may be clarified or completed:

Please change the unit “kms” for km; What kind of “tridimensional geomats” (line 169) and “synthetic geotextile” (Fig. 5) were used? I think that the authors may highlight some information regarding two other important issues: construction costs and maintenance costs (if possible, including a comparison with other “traditional” solutions).

Author Response

Reviewer #3:

Congratulations to authors. Very interesting paper regarding a current theme: an ecological way to minimize some consequences of gravel extraction in rivers.

In general the paper is well presented but some information may be clarified or completed:

1-Please change the unit “kms” for km;

It has been replaced.

2-What kind of “tridimensional geomats” (line 169) and “synthetic geotextile” (Fig. 5) were used?

We call tridimensional geomat to tridimensional geosynthetcscused at the soil-geotextile for the reinforcement interface, this is, a geotextile is placed between synthetic membranes for protective proposes, whereas synthetic geotextile is just a synthetic membrane.

3-I think that the authors may highlight some information regarding two other important issues: construction costs and maintenance costs (if possible, including a comparison with other “traditional” solutions).

This aspect it is not contemplated, not because it is not important, but the number of these type of rehabilitation procedures is so scarce in Portugal  that no comparisons are possible at this stage.

Reviewer 4 Report

The authors illustrate a restoration project, integrating hard structures with soft ecological engineering procedures (re-vegetation of the river bed). These applications involve two distinct segments of a Portugal’s river, affected by a relevant deepening of the river bed. The methodology is appropriate and the manuscript is technically sound, being also an interesting case study from scientific point of view. It is well written and organized, the reader easily interprets text, tables and figures; the abstract and keywords are appropriate.

Comments:

1) Fig. 3 is not clear, it needs a legend.

2) A broad discussion is needed on the fact that “a more complete study at a larger regional scale is needed… in order to include the appropriate management that may contribute to overcome the deficit of sediments in the estuarine area". In fact, the disappointing results of the illustrated arrangement principally depend on the lack of sediment, trapped upstream.

3) It seems also necessary to debate the role of instream vegetation: on one side it is an important water storage when flooding events occur and on the other, it could cause an increase of water level, increasing the hydraulic risk.

Author Response

Reviewer #4:

The authors illustrate a restoration project, integrating hard structures with soft ecological engineering procedures (re-vegetation of the river bed). These applications involve two distinct segments of a Portugal’s river, affected by a relevant deepening of the river bed. The methodology is appropriate and the manuscript is technically sound, being also an interesting case study from scientific point of view. It is well written and organized, the reader easily interprets text, tables and figures; the abstract and keywords are appropriate.

Comments:

1) Fig. 3 is not clear, it needs a legend.

Figure 3 legend has been added.

2) A broad discussion is needed on the fact that “a more complete study at a larger regional scale is needed… in order to include the appropriate management that may contribute to overcome the deficit of sediments in the estuarine area". In fact, the disappointing results of the illustrated arrangement principally depend on the lack of sediment, trapped upstream.

We believe now that in the introduction and discussion we call the attention now for the complexity of disturbance sources and the need for an integrate management.

3) It seems also necessary to debate the role of instream vegetation: on one side it is an important water storage when flooding events occur and on the other, it could cause an increase of water level, increasing the hydraulic risk.

Thre is practically no instream vegetation, this is an important biomass of macrophytes that may create resistence to flow and that may increase flooding area; on the other hand, salt marshes surround the the river channel and they may contribute to dissipate energy and to vanish the impact of peak flows. Moreover, are important wetlands that may store excesso of water in high flow periods.

Round 2

Reviewer 1 Report

The authors have revised the paper somewhat to help clarify the main objectives of the study and what analyses were actually conducted to assess the success of the two bank stabilization projects. This has improved the manuscript; however, there are still several outstanding issues that need to be addressed

General comments:

The revisions to the Introduction and re-organization of the Methods and Results sections are helpful. However, there is still insufficient description of the HEC-RAS modeling. I realize the authors are trying to limit the length of the manuscript (although Water has no length limit for articles). Still, even adding the few brief sentences that the authors included in the response to my earlier review would be helpful. For example, they chose the flow magnitudes to correspond to two frequent events (2.33 and 5 year) and one extreme event (100 year). Also, include how the authors selected the Manning’s n values to use and include in Table 3 (or elsewhere) what the pre- and post-restoration Manning’s n values were.

In the revised manuscript, the authors make a useful point that these projects can be used to determine the most appropriate restoration strategies for applying elsewhere in the estuary (Lines 84-86). However, this should be brought up again in the discussion to assess how well these types of bank stabilization projects might work based on the performance of these two projects. What lessons were learned that can be applied to future projects? This is partially addressed in Lines 300-322, but additional text on what specific improvements could be made (e.g. importing fine sediment at the bank toe to encourage plant survival?) would be helpful.

Many of my previous review comments were not addressed in the revised manuscript, even though the authors claim to have made edits. Many of these are relatively minor suggestions about clarifying specific sentences. However, I still believe these sections are unclear and the writing can be improved. Also, for other comments, the authors provided a helpful clarification in their response, but did not make any changes to the manuscript. I believe these areas should be clarified within the manuscript as readers may have similar questions as I do.

I have provided some additional comments and suggestions for edits in the attached pdf.

Author Response

Dear Reviewer,

Included is the revised version a manuscript entitled “Combination of ecological engineering procedures applied to morphological stabilization of estuarine banks after dredging”, for eventual publication in the Water journal. Authors declare no conflict of interest.

We attached a pdf document with responded thoroughly to all comments and suggestions, of the minor revision, and believe the revised version has benefitted greatly from them.
